# Antibody Response Following the Intranasal Administration of SARS-CoV-2 Spike Protein-CpG Oligonucleotide Vaccine

**DOI:** 10.3390/vaccines12010005

**Published:** 2023-12-20

**Authors:** Kentaro Muranishi, Mao Kinoshita, Keita Inoue, Junya Ohara, Toshihito Mihara, Kazuki Sudo, Ken J. Ishii, Teiji Sawa, Hiroyasu Ishikura

**Affiliations:** 1Department of Emergency and Critical Care Medicine, Faculty of Medicine, Fukuoka University, Fukuoka 814-0133, Japan; muranishi@adm.fukuoka-u.ac.jp (K.M.); ishikurah@fukuoka-u.ac.jp (H.I.); 2Department of Anesthesiology, Graduate School of Medical Science, Kyoto Prefectural University of Medicine, Kyoto 602-8566, Japan; keitaino@koto.kpu-m.ac.jp (K.I.); j-ohara@koto.kpu-m.ac.jp (J.O.); tmihara@koto.kpu-m.ac.jp (T.M.); a080025@koto.kpu-m.ac.jp (K.S.); anesth@koto.kpu-m.ac.jp (T.S.); 3Division of Vaccine Science, Department of Microbiology and Immunology, The Institute of Medical Science, The University of Tokyo, Tokyo 108-8639, Japan; kenishii@ims.u-tokyo.ac.jp

**Keywords:** SARS-CoV-2, intranasal administration, mucosal immunity, recombinant protein, CpG-deoxyoligonucleotide

## Abstract

The new coronavirus infection causes severe respiratory failure following respiratory tract infection with severe acute respiratory syndrome-related coronavirus (SARS-CoV-2). All currently approved vaccines are administered intramuscularly; however, intranasal administration enhances mucosal immunity, facilitating the production of a less invasive vaccine with fewer adverse events. Herein, a recombinant vaccine combining the SARS-CoV-2 spike protein receptor-binding domain (RBD), or S1 protein, with CpG-deoxyoligonucleotide (ODN) or aluminum hydroxide (alum) adjuvants was administered intranasally or subcutaneously to mice. Serum-specific IgG titers, IgA titers in the alveolar lavage fluid, and neutralizing antibody titers were analyzed. The nasal administration of RBD protein did not increase serum IgG or IgA titers in the alveolar lavage fluid. However, a significant increase in serum IgG was observed in the intranasal group administered with S1 protein with CpG-ODN and the subcutaneous group administered with S1 protein with alum. The IgA and IgG levels increased significantly in the alveolar lavage fluid only after the intranasal administration of the S1 protein with CpG-ODN. The neutralizing antibody titers in serum and bronchoalveolar lavage were significantly higher in the intranasal S1-CpG group than in every other group. Hence, the nasal administration of the S1 protein vaccine with CpG adjuvant might represent an effective vaccine candidate.

## 1. Introduction

Coronavirus disease (COVID-19) remains a considerable global threat and results from infection with severe acute respiratory syndrome coronavirus 2 (SARS-CoV-2), potentially causing severe respiratory failure [1]. Since the spring of 2020, COVID-19 has remained highly infectious, with SARS-CoV-2 continuously evolving to create several genomically unique variants with varying levels of severity and immune evasion [2]. The mechanism of cell fusion involves the receptor-binding domain (RBD) of the viral surface antigen protein, spike protein (S protein), binding to the angiotensin-converting enzyme 2 (ACE2) receptor in human airway epithelial cells [3]. When the SARS-CoV-2 antigen reaches the mucous membrane, it is taken up by M cells, resulting in nasopharynx-associated lymphoid tissues generating IgA-producing B cells and the subsequent production of mucosal-secreted IgA. Subsequently, the induction of systemic humoral and cell-mediated immunity can prevent pneumonia and the infection of essential organs, eliminate viruses in the upper and lower respiratory tracts, acquire long-term immunity, and prevent the spread of the disease [4]. Accordingly, mucosal immunity is an important factor in preventing and treating infections [5]. However, all authorized available vaccines are administered intramuscularly with weak mucosal immunity induction [6]. Different types of vaccines, including viral vector, mRNA, and protein subunit vaccines, have been investigated to develop safe and effective vaccines against SARS-CoV-2, some of which have been approved for clinical use [6]. Moreover, an intranasally (i.n.) administered adenoviral vector vaccine expressing the SARS-CoV-2 spike protein has demonstrated protective effects in mice and rhesus monkeys [7]. However, little is known regarding the immunogenicity of intranasal immunization with SARS-CoV-2 subunit-based vaccines. Currently, licensed vaccines for SARS-CoV-2 are prepared as subcutaneous (s.c.) inoculations; however, given the importance of mucosal immunity for combatting respiratory viruses, intranasal formulations are being investigated for various respiratory viruses. To date, intranasal attenuated live influenza vaccines have become an important aspect of the influenza vaccination strategy, while clinical trials are underway for an intranasal pertussis vaccine [8]. Therefore, generating intranasally administered vaccines that enhance the immunity of the respiratory tract mucosa while being less invasive with fewer side effects is imperative.

In order to elicit mucosal immunity using the molecular components of pathogens as vaccine antigens, it is necessary to also select safe and effective adjuvants. Aluminum hydroxide (alum), a typical clinical adjuvant, is cytotoxic and causes cell necrosis [8]. Moreover, we recently found that a DNA oligonucleotide with a CpG sequence (CpG-ODN) stimulates Toll-like receptor-9 on mucosal cells, serving as an effective mucosal adjuvant in the development of an intranasal vaccine against *Pseudomonas aeruginosa* [9].

In this study, CpG and alum were selected as two clinically low-risk adjuvants to enhance the immunogenicity of the SARS-CoV-2 spike protein. We hypothesized that combining the SARS-CoV-2 spike protein antigen and CpG-ODN adjuvant will induce effective mucosal immunity. In order to test this hypothesis, we combined recombinant RBD protein, or S1 protein, as the vaccine antigen with CpG ODN or alum as the adjuvant and administered the vaccine intranasally or subcutaneously to mice. We then examined the changes in the specific and neutralizing antibody titers of the spike protein in the serum and bronchoalveolar lavage (BAL) fluid.

## 2. Materials and Methods

### 2.1. Recombinant Protein and Adjuvants

The RBD and S1 proteins were purchased from Sino Biological (SARS-CoV-2 Spike Protein [RBD, His Tag], #40592-V08H, SARS-CoV-2 Spike S1-His Recombinant Protein [HPLC-verified], #40591-V08H, Beijing, China). Aluminum hydroxide gel was purchased from InvivoGen (Alhydrogel^®^ adjuvant 2%, #vac-alu-250, San Diego, CA, USA). CpG K type (K3) ODN (5′-ATC GAC TCT GGA GGG TTC TC-3′) was prepared by GeneDesign (Ibaraki, Osaka, Japan) and dissolved in saline to 1 mg/mL.

### 2.2. Vaccine Efficacy Study

The protocols for all animal experiments were approved by the Animal Research Committee of the Kyoto Prefectural University of Medicine (approval numbers: M2020-535 and M2021-337). Male Institute of Cancer Research (ICR) mice (5 weeks old; body weight, 25 g) were certified pathogen-free and purchased from Shimizu Laboratory Supplies, Co., Ltd. (Kyoto, Japan). Mice were housed in groups of six per cage with filtertops under pathogen-free conditions. Recombinant RBD protein (0.5 mg/mL) and S1 protein (0.5 mg/mL) were combined with CpG-ODN (hereafter CpG) or alum. A set of experiments was conducted independently on different days to ensure the reproducibility of the results; each batch had three vaccination groups with 10 mice per group.

#### 2.2.1. Characterization of RBD, Adjuvant, Route of Administration Humoral Immune Response

Total dose volumes of 30 µL and 200 µL were administered nasally or subcutaneously, respectively. Mice received subcutaneous injections in the back or, for intranasal administration, were placed under general anesthesia with sevoflurane, and 15 µL of the vaccine was administered into the left and right nasal cavities (a total of 30 µL) using a pipette. RBD (10 µg)-CpG (1000 µg) or CpG (1000 µg) alone was nasally administered, as previously described [10,11]. RBD (10 µg)-alum (1000 µg) or alum (1000 µg) alone was subcutaneously administered, as previously described [12]. In the first experiment, the RBD-CpG, RBD-alum, CpG alone, and alum alone groups were administered on days 0 and 28. The mice were euthanized on day 56 while collecting blood and bronchoalveolar lavage (BAL) fluid.

#### 2.2.2. Characterization of RBD-CpG Intranasal Humoral Immune Response

The vaccine was administered via the nasal route. The total dose volume was 30 µL for nasal administration. For intranasal administration, the mice were placed under general anesthesia with sevoflurane, and 15 µL each from the left and right nasal cavities (a total of 30 µL) were instilled intranasally using a pipette. The RBD-CpG was administered intranasally. In the RBD (10 µg)-CpG (1000 µg) group, the vaccine solution was administered on days 0, 7, 14, 21, 35, and 49, and blood and BAL fluid were collected on day 56.

#### 2.2.3. Characterization of S1, Adjuvant, Route of Administration Humoral Immune Response

The vaccine was administered nasally and subcutaneously. The total dose volume was 30 µL for nasal administration and 200 µL for subcutaneous administration. Subcutaneous injections were administered to the backs of mice. For intranasal administration, the mice were placed under general anesthesia with sevoflurane, and 15 µL from the left and right nasal cavities (a total of 30 µL) each were instilled intranasally using a pipette. We compared the effects of S1 (10 µg)-CpG (1000 µg), S1 (10 µg)-alum (1000 µg), S1 (10 µg) alone (nasal and subcutaneous), and saline (nasal and subcutaneous) administration. In the intranasal administration group, the vaccine was administered on days 0, 7, 14, and 21, and blood and BAL fluid were collected on day 42. In the subcutaneously administered group, the vaccine was injected on days 0 and 28, and blood and BAL fluid were collected on day 56.

### 2.3. Serum Collection and Bronchoalveolar Lavage (BAL) Fluid Collection

Blood was collected from the tail vein and carotid artery, centrifuged at 10,000 rpm for 10 min, and the serum was stored at −80 °C.

After euthanasia with pentobarbital or sevoflurane, a tracheotomy was performed; a total of 2–3 mL of phosphate-buffered saline (PBS) was injected into the lungs with a catheter, and the BAL fluid was collected while vibrating with a vibration machine. The recovery rate was approximately 50–70%. Following centrifugation at 1000 rpm for 10 min, the supernatant was mixed with 50% glycerol and stored at −80 °C.

### 2.4. SARS-CoV-2 Specific Enzyme-Linked Immunosorbent Assay (ELISA)

RBD protein (Abcam, recombinant human coronavirus SARS-CoV-2 Spike Glycoprotein RBD [Active] 50 µg, #ab273065, Cambridge, UK) or S1 protein (Sino Biologicals, SARS-CoV-2 Spike S1-His Recombinant Protein [HPLC-verified], #40591-V08H) was diluted with a coating solution (0.1 mM NaHCO_3_, pH 9.6) to a concentration of 2 µg/mL. Microtiter plates (430341, Nunc C96 Maxisorp; Thermo Fisher Scientific, Waltham, MA, USA) were coated for 2 h at 4 °C with each diluted antigen. Subsequently, 200 µL/well of blocking solution was added and incubated overnight at 37 °C or for 2 h or four times. A 1000-fold dilution of mouse serum or BAL fluid was used as the primary antibodies and added to the plates (100 μL/well) and incubated overnight at 4 °C. Peroxidase-labeled anti-mouse IgG (Sigma-Aldrich, anti-mouse IgG [whole molecule] produced in goat-affinity isolated, buffered aqueous solution, #A4416, St. Louis, MO, USA) and anti-mouse IgA (Abcam, Waltham, MA, USA, goat anti-mouse IgA alpha chain [HRP], #ab97235) were applied at 1:60,000 for 1 h at 37 °C. After six washes, the plates were incubated with 2,2′-azino-bis (3-ethylbenzthiazoline-6-sulfonic acid) (A3219; Sigma-Aldrich) at room temperature for 30 min. Next, 0.5 M H_2_SO_4_ at 100 µL was added per well, and the optical density (OD) was measured at 450 nm using a microplate reader (MTP-880Lab; Corona Electric, Hitachinaka, Japan).

### 2.5. Anti-SARS-CoV-2 Neutralization Titers

Neutralizing antibodies were purchased from AdipoGen Life Science (SARS-CoV-2 Neutralization Antibodies Detection Kit, #AG-48B-0002-KI01, San Diego, CA, USA). The antigens were obtained from SARS-CoV-2 (Wuhan-Hu-1). The RBD protein was preliminarily immobilized on the 96-well plate provided in the kit, and 100 µL/well of a 10-fold diluted serum or undiluted BAL samples were added and incubated at 37 °C for 1 h. Subsequently, 100 µL/well of ACE2 (human)-HRP was added and reacted for 1 h at 37 °C; a total of 100 µL/well of 3,3′,5,5′-tetramethylbenzidine (TMB) solution was added and incubated for 10 min in the dark before adding 100 µL/well of stop solution and measuring the absorbance at OD_450_ nm using a microplate reader (MTP-880Lab; Corona Electric, Hitachinaka, Japan).

### 2.6. Statistical Analysis

One-way ANOVA and Tukey’s multiple comparison tests were used to compare anti-S1 IgG, anti-S1 IgA, anti-RBD IgG, and anti-RBD IgA titers. A *p* < 0.05 was considered significant. Statistical analyses were performed using Prism 8 software (GraphPad Software, La Jolla, CA, USA). Data are presented as mean, standard deviation (SD), or median values.

## 3. Results

### 3.1. Intranasal or Subcutaneous Administration of Adjuvanted RBD and S1 Recombinant Protein Induces Humoral Systemic Responses in Mice

#### 3.1.1. Effect of RBD, Adjuvant, and Route of Administration on Humoral Immune Response

On days 0 and 28, the vaccines containing RBD (10 µg)-CpG (1000 µg) or CpG (1000 µg) alone were intranasally administered, whereas RBD (10 µg)-alum (1000 µg) or alum (1000 µg) alone were subcutaneously administered (*n* = 10/group). On day 56, the serum and IgG BAL antibody titers and the IgA BAL antibody titers were measured (Figure 1). A significant increase in the IgG antibody titer was observed in the serum against RBD only in the RBD-alum subcutaneous group (Figure 1a). However, no significant increase was observed in the BAL IgA antibody titers in any group (Figure 1b).

#### 3.1.2. Characterization of the RBD-CpG Intranasal Humoral Immune Response

Since no increase was observed in the antibody titers after intranasal administration in the RBD-CpG intranasal administration group, we increased the number of intranasal administrations of RBD (10 µg with CpG 1000 µg) from two to six times in 10 mice. However, no significant increase in serum or BAL IgG antibody titers was observed over the 56 days (Figure 2).

#### 3.1.3. Effect of S1, Adjuvant, and Route of Administration on Humoral Immune Response

In order to test the hypothesis that mucosal immunity is induced using S1 as an antigen and CpG as an adjuvant, S1 (10 µg)-CpG (1000 µg), S1 (10 µg)-alum (1000 µg), or S1 (10 µg) alone were administered intranasally or subcutaneously (*n* = 10/group). Saline was administered four times intranasally or twice subcutaneously to three mice, each as a sham control. The serum and BAL antibody titers were measured on days 42 and 56 for the intranasal and subcutaneous administration groups, respectively (Figure 3 and Figure 4).

Intranasal administration of S1-CpG and subcutaneous administration of S1-alum significantly increased the serum IgG antibody titers against the S1 antigen (Figure 3a) and RBD antigen of the spike protein S1 (Figure 3b). The BAL IgA antibody titers against the S1 antigen (Figure 4a) and RBD antigen (Figure 4b) significantly increased following the intranasal administration of S1-CpG. Finally, the BAL IgG antibody titers against the S1 antigen (Figure 4c) and RBD antigen (Figure 4d) significantly elevated following the intranasal administration of S1-CpG.

### 3.2. Anti-SARS-CoV-2 Neutralization Titers

The neutralizing antibody titers against the original (Wuhan-Hu-1) SARS-CoV-2 virus were assessed to investigate the presence of the neutralizing antibody fraction. A significant increase was observed in the number of neutralizing antibodies in the serum of the S1-CpG and S1-Alum groups (Figure 5a). The neutralizing antibody titers in the serum and BAL were significantly higher following intranasal S1-CpG administration than in every other group (Figure 5b).

## 4. Discussion

Currently, all SARS-CoV-2 vaccines approved for clinical use are administered intramuscularly, primarily inducing a serum IgG immune response, not mucosal secretory IgA antibodies [11,13]. Therefore, although intramuscular vaccines prevent severe disease in infected patients, they cannot halt the spread of the contagion from infected patients to others, and the vaccinated individuals may become asymptomatic carriers [14,15]. IgA antibodies are detected earlier than IgG antibodies in SARS-CoV-2-infected individuals and persist for a long time [16]. In this study, the intranasal administration of S1-CpG increased serum IgG and BAL IgA antibody titers against S1 and RBD, resulting in a significant increase in neutralizing antibodies in serum and BAL. This was not achieved by the intranasal administration of RBD-CpG. Nasopharynx mucosal-secreted IgA antibodies are approximately 7.5-fold more potent than serum IgG antibodies [17]. Furthermore, the T cells and B cells activated by mucosal secretory IgA reportedly retain the immune memory and provide long-term systemic and local mucosal immunity [18]. The nasal administration of the S1 protein vaccine with CpG as an adjuvant is effective and may provide long-term systemic and local mucosal immunity.

In ferret experiments, a single intranasal administration of a full-length S protein adenoviral vector increased specific mucosal secretory IgA and serum IgG titers; however, intramuscular administration did not increase mucosal IgA titers [19]. The intranasal administration of S protein adenovirus vector vaccines to rhesus monkeys results in the production of neutralizing antibodies, cell-mediated immune responses, and suppression of viral infection [20]. Moreover, ChAd-SARS-CoV-2—an adenoviral vector vaccine—significantly decreased the viral load in the upper respiratory tract when administered intranasally compared to subcutaneously, thus reducing viral replication and transmission [19,21]. In the macaque model, our previous research has demonstrated that administering a third or fourth dose of an adjuvanted subunit vaccine provides complete protection against viral challenge. Therefore, the experimental design for this study is based on administering a weekly booster dose. Hence, intranasal administration improves the induction of mucosal immunity via IgA, and, as shown in the current study, this can also induce systemic immunity. However, many reports are based on viral vector vaccines, with few studies on the intranasal administration of recombinant proteins.

Compared with mRNA vaccines—primary SARS-CoV-2 vaccines used globally—recombinant protein vaccines are more stable and less dependent on the cold chain, making them easier to manufacture and distribute worldwide [22,23]. However, recombinant proteins exhibit poor immunogenicity and require appropriate adjuvants to induce immunity [24]. Recent studies have revealed that the intranasal administration of a recombinant RBD protein using the cationic nanocarrier polyethyleneimine (PEI) as an adjuvant significantly increased humoral and cell-mediated immunity in mice [25]. Additionally, the nasal administration of a recombinant protein using alum as an adjuvant to the RBD protein in mice significantly increased IgG levels compared to intradermal and intramuscular administration. Furthermore, the induction of cell-mediated immunity, including IFNγ and IL-4 production, was reported [26]. Hence, these results have demonstrated the usefulness of intranasal administration.

In this study, alum was used as an adjuvant for subcutaneous administration and was compared with CpG-ODNs. Alum has been used as an adjuvant in the diphtheria-tetanus-pertussis and polio vaccines [24,27]. It is currently approved for clinical use by the US Food and Drug Administration (FDA). However, the mechanism of alum’s immunostimulatory action is not fully understood [28,29,30]. Recent studies have suggested that aluminum-induced cell death and the subsequent release of host cell DNA serve as potent endogenous immunostimulatory signals [9]. In contrast, CpG-ODNs are short, single-stranded, synthetic DNA fragments that contain unmethylated CpG dinucleotides in specific sequences. These sequences include CpG motifs, which are common in most bacterial, viral, and invertebrate genomes [31]. CpG ODN mimics the immunostimulatory effects of bacterial DNA and activates cells that express Toll-like receptor 9, such as plasmacytoid dendritic cells and B cells [32]. Once activated, these cells produce inflammatory cytokines and activate natural killer cells, monocytes, and neutrophils, ultimately leading to the activation of Th1 immune responses [31,33]. Recently, CpG ODN was used as an intranasal adjuvant. Within the influenza vaccine and tetanus toxoid vaccine, CpG ODN activates immunity upon mucosal administration [34,35]. Moreover, a PcrV-CpG vaccine, which is used to prevent *Pseudomonas aeruginosa* lung infection, induced anti-PcrV IgA titers in BAL and lung homogenates and anti-PcrV IgG titers in serum that were significantly higher than in the PcrV-alum group [10]. Hence, PcrV-CpG induces mucosal and systemic immune responses, acting as an appropriate adjuvant to enhance the immunogenicity of the vaccine. Although intranasal CpG ODN induces mucosal immunity, few studies have assessed its efficacy in the context of SARS-CoV-2 vaccines. Nevertheless, Yongjun et al. administered the S1 protein and alum subcutaneously to rhesus monkeys, followed by the intranasal administration of a recombinant protein vaccine containing the S1 protein as a booster with CpG ODN, PolyI:C, and IL-15 as adjuvants to increase the BAL IgA titer and decrease viral RNA [6]. Few studies have reported the intranasal administration of S1 protein vaccines with CpG ODN as adjuvants, highlighting the potential usefulness of CpG ODN in inducing mucosal immunity. Most adverse events associated with the subunit recombinant spike protein vaccine combined with the Advax-CpG55.2™ adjuvant were mild, with complete recovery occurring relatively quickly [36].

In the present study, the RBD protein was used as an antigen. The RBD is an attractive antigen for subunit vaccines as it mediates the entry of viruses into host cells and is a target for neutralizing antibodies. However, intranasal administration did not increase serum IgG or BAL IgA titers; this agrees with the poor immunogenicity previously reported for the RBD [11]. Given their small size, RBDs are not readily taken up by the lymphatic system, limiting their interaction with important immune antigens [11] and resulting in their rapid dispersion from the injection site and elimination, often showing poor pharmacokinetics [13]. In the current study, the same doses of S1 protein and RBD were administered. However, the antibody response varied depending on the size of the molecule. Intranasal vaccination is a less invasive method compared to skin penetration injections, and it can trigger both systemic and mucosal immunity [37,38]. The mucosa of the nasopharyngeal duct contains plasmacytoid dendritic cells (pDCs) that have antigen-presenting capacity. When antigens reach pDCs, antigen-specific CD4+ helper T cells are activated, and they stimulate immature B cells to differentiate into IgA-producing plasma cells. These plasma cells secrete IgA (S-IgA) at effector sites [39]. S-IgA binds to toxins, bacteria, and viruses, neutralizing their activity and preventing their entry into the mucosal epithelium. Despite being an effective strategy for inducing an immune response, intranasal vaccination may lead to mucosal tolerance if the antigen has weak immunogenicity [33]. Therefore, to effectively induce antigen-specific mucosal immunity, proper selection of the adjuvant and mucosal route of administration is imperative.

However, RBD reportedly causes an antibody response equivalent to full-length S, S1, and S2 [6], with the intranasal administration of RBD-alum increasing antibody titers [24]. Therefore, the precise reason why RBD administration failed to induce an increase in antibody titers is unknown. Nevertheless, an important finding of this study is that only the intranasal administration of S1-CpG increased the IgA titer. Furthermore, when the neutralizing antibody fraction of the initial Wuhan-hu-1 strain was examined, the fraction from the nasal administration of S1-CpG in serum was equivalent to that from the subcutaneous administration of S1-alum; only the fraction of intranasally administered S1-CpG was higher in BAL. Spike-LP-GMP administered intranasally twice or administered as a booster intranasally once after parenteral immunization induces CD4+ and CD8+ TRM cells in the nose and lungs, including the systemic and local IgG and IgA and Tfh cells necessary for antibody production by B cells [40]. Thus, the superior immunogenicity of the spike protein may also be attributed to its trimeric structure, which is considerably more stable than the monomeric form of RBD. Repeated trimeric antigen presentation of the spike protein may increase cross-linking to B cell receptors and induce greater B cell activation and IgA responses [41].

When compared to subcutaneously administered vaccines, nasal administration more readily activates mucosal immune responses, does not require a sterile environment, can be administered safely to children of all ages, circumvents the risk associated with inappropriate needle use or risk of infectious disease transmission, and the boosters can be easily administered by patients [42].

Certain limitations were noted in this study. First, we evaluated humoral immunity via the quantification of antibody titers but did not analyze cell-mediated immunity, which is also crucial in SARS-CoV-2 immunity. Therefore, assessing cell-mediated immunity, such as cytokine production, is necessary. Second, the prevention of infection after immunization is critical; however, this was not assessed in this study. Assuming that these issues can be resolved, the minimally invasive and highly convenient intranasal S1 protein with CpG ODN vaccine will become an attractive option for boosting protective immunity against the virulent SARS-CoV-2.

## 5. Conclusions

We conclude that the RBD portion of the S1 protein alone is insufficient to induce immunity regardless of the route of administration, with alum or CpG as adjuvants. However, the nasal administration of the S1 protein vaccine with CpG as an adjuvant is efficacious, representing a potential vaccine candidate.

## Figures and Tables

**Figure 1 vaccines-12-00005-f001:**
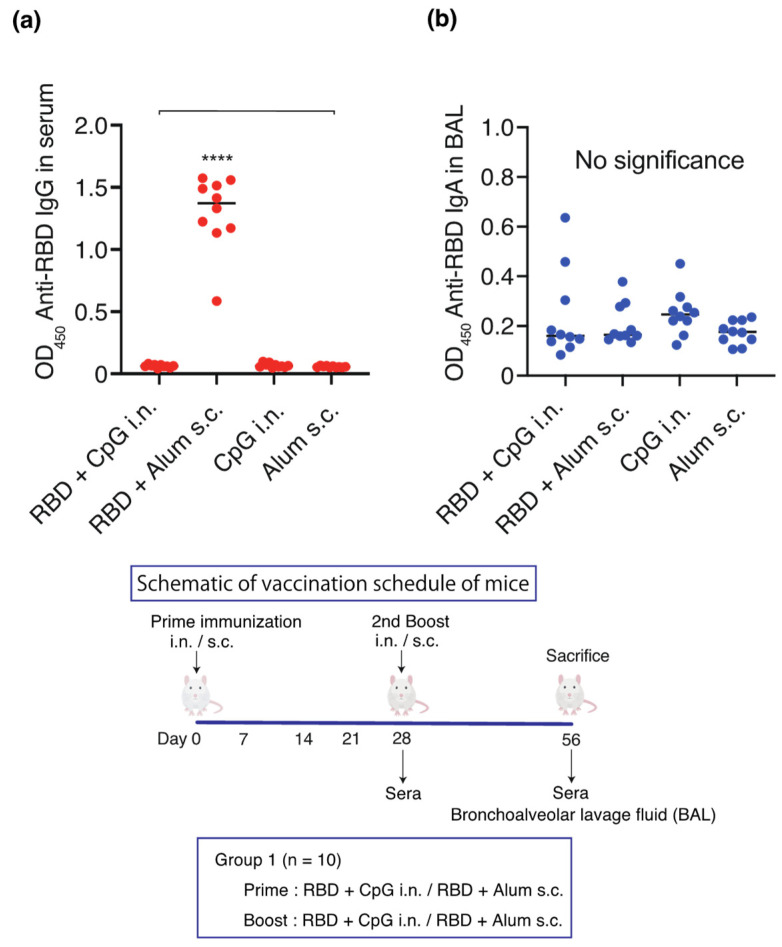
Schematic of the vaccination schedule. Mice received vaccines or control solutions intranasally or subcutaneously on days 0 and 28. On day 56, the vaccinated mice were euthanized, and blood and bronchoalveolar lavage (BAL) fluids were collected for anti-RBD titer measurement: anti-RBD IgG titers in the serum and anti-RBD IgA titers in the BAL. (**a**) Anti-RBD IgG titers in diluted 1:1000 serum. (**b**) Anti-RBD IgA titers in BAL. Titers were measured 56 days after the initial administration of RBD-CpG, RBD-alum, CpG alone, or alum alone. The closed circles indicate individual data points. Data are shown as median (solid central bar in the box) and quartiles (boxes); **** *p* < 0.0001 compared to all other groups. BAL, bronchoalveolar lavage; OD_450_, optical density measurements using a microplate reader with a 450 nm filter. RBD, receptor-binding domain; i.n., intranasal; s.c., subcutaneous.

**Figure 2 vaccines-12-00005-f002:**
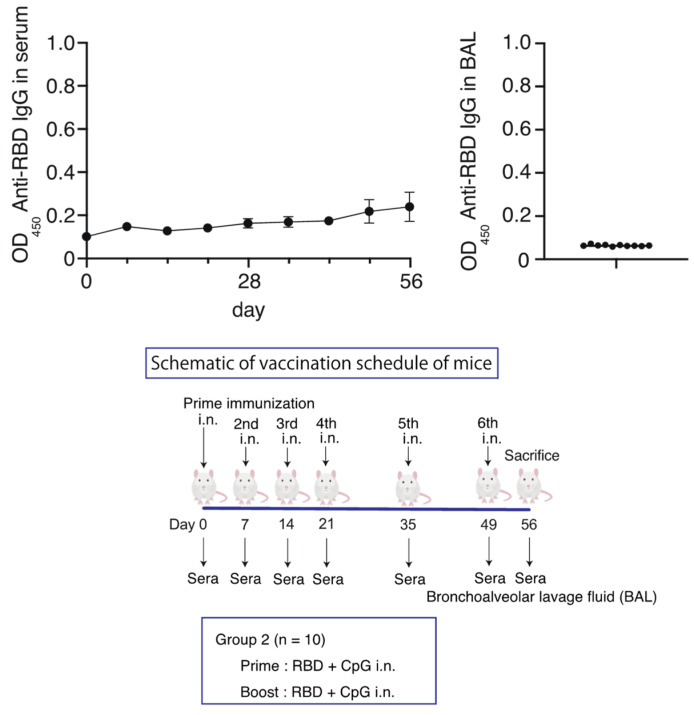
Schematic of vaccination schedule for the RBD-CpG intranasal administration group. Mice received vaccines or control solutions four times on days 0, 7, 14, 35, and 49. On day 56, the vaccinated mice were euthanized, and blood and BAL fluids were collected for anti-RBD titer measurement. Anti-RBD IgG titers in serum and BAL from the vaccinated mice. The circles and the bars are shown as median ± SD. OD_450_, optical density at 450 nm; RBD, receptor-binding domain. i.n., intranasal; s.c., subcutaneous.

**Figure 3 vaccines-12-00005-f003:**
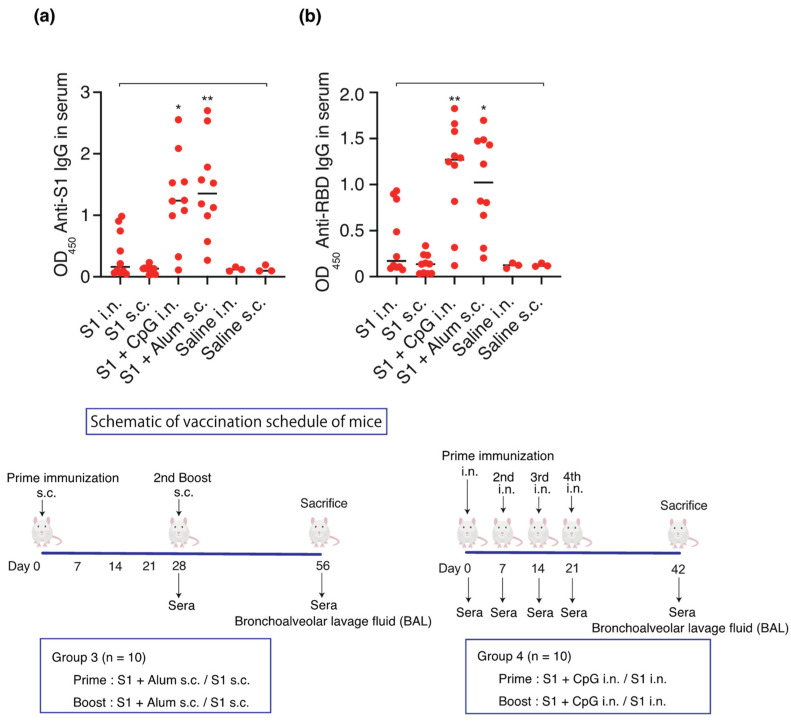
Schematic of vaccination schedule: S1-CpG, S1-alum, S1 intranasal, S1 subcutaneous, saline intranasal, or subcutaneous administration. Mice received vaccines or control solutions intranasally four times on days 0, 7, 14, and 21. On day 42, the vaccinated mice were euthanized, and blood was collected for anti-RBD/S1 titer measurement. Mice received vaccines or control solutions subcutaneously twice on days 0 and 28. On day 56, the vaccinated mice were euthanized, and blood was collected for anti-RBD/S1 titer measurement. (**a**) Anti-S1 IgG titers in 1:1000 diluted serum. (**b**) Anti-RBD IgG titers in 1:1000 diluted serum. The closed circles show the individual data. The bars represent the median. * *p* < 0.05, ** *p* < 0.01 compared to all other groups. OD_450_, optical density measurements at 450 nm. RBD, receptor-binding domain; i.n., intranasal; s.c., subcutaneous. Samples with OD values ≥ 2.5 are deemed unaffected in relation to statistical outcomes.

**Figure 4 vaccines-12-00005-f004:**
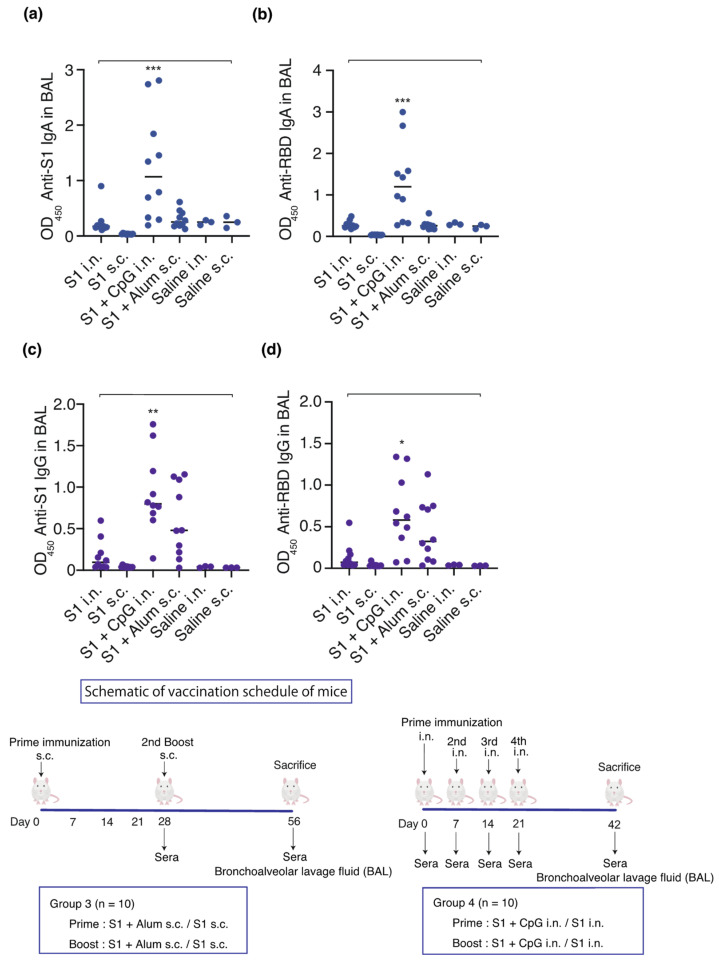
Effect of S1, adjuvant, and route of administration on the humoral immune response. Schematic of vaccination schedule for mice: S1-CpG, S1-alum, S1 intranasal, S1 subcutaneous, saline intranasal, or subcutaneous administration group. Mice received vaccines or control solutions intranasally four times on days 0, 7, 14, and 21. On day 42, the vaccinated mice were euthanized, and BAL fluids were collected for anti-RBD/S1 titer measurement. Mice received vaccines or control solutions subcutaneously twice on days 0 and 28. On day 56, the vaccinated mice were euthanized, and BAL fluids were collected for anti-RBD/S1 titer measurement. Anti-S1 IgA and anti-RBD IgA titers in BAL from the vaccinated mice. (**a**) Anti-S1 IgA titers in BAL. (**b**) Anti-RBD IgA titers in BAL. (**c**) Anti-S1 IgG titers in BAL. (**d**) Anti-RBD IgG titers in BAL. The closed circles represent individual data; bars represent the median. * *p* < 0.05, ** *p* < 0.01, and *** *p* < 0.001 compared to all other groups. BAL, bronchoalveolar lavage; OD_450_, optical density measurements at 450 nm; RBD, receptor-binding domain. i.n., intranasal; s.c., subcutaneous. Samples with OD values ≥ 2.5 are deemed unaffected in relation to statistical outcomes.

**Figure 5 vaccines-12-00005-f005:**
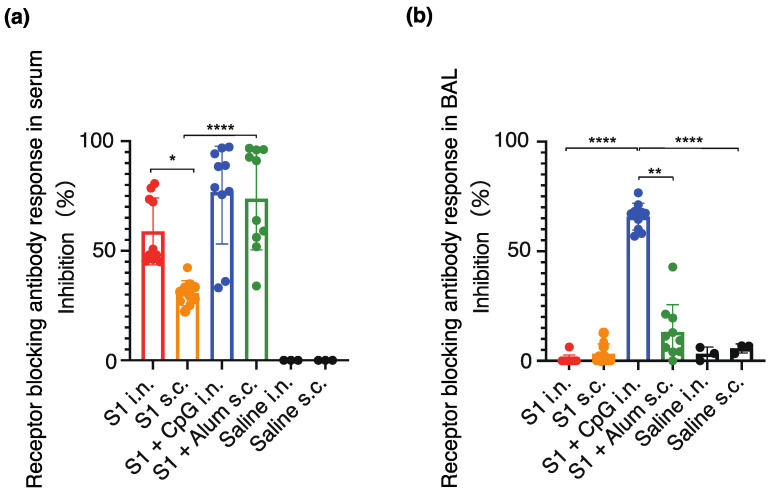
Serum (**a**) and BAL (**b**) receptor blocking antibody responses against SARS-CoV-2 (Wuhan-Hu-1). The closed circles represent individual data; boxes and bars represent the mean ± SD. * *p* < 0.01 between S1 intranasal and S1 subcutaneous; ** *p* < 0.01 between S1 + CpG intranasal and S1 + alum subcutaneous; **** *p* < 0.0001 compared to all other groups. i.n., intranasal; s.c., subcutaneous.

## Data Availability

The datasets generated and/or analyzed in this study are available from the corresponding author upon reasonable request.

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
