# Peer review of "Antibody Response Following the Intranasal Administration of SARS-CoV-2 Spike Protein-CpG Oligonucleotide Vaccine"

_vaccines, 2023, doi:10.3390/vaccines12010005_

Round 1
Reviewer 1 Report (Previous Reviewer 2)
Comments and Suggestions for Authors
The manuscript has significantly improved with the inclusion of additional experiments and controls. Before publication, I recommended to the author that they modify the Y-axis title of Figure 5 to something like "Receptor blocking antibody response in serum" or a similar description. This adjustment was suggested because the method utilized in Figure 5 did not align with a canonical neutralization assay, despite initially being labeled as "neutralizing antibody."
Comments on the Quality of English LanguageMinor editing of English language required
Author Response
We thank the reviewer for their valuable comments and suggestions. In Figure 5, we have made a correction by adding "Receptor blocking antibody response in serum"and "Receptor blocking antibody response in BAL" as the names for the y-axis.
Reviewer 2 Report (New Reviewer)
Comments and Suggestions for Authors
The manuscript describes a recombinant SARS-CoV-2 spike protein vaccine containing CpG-ODN and given intranasally that induces a strong mucosal immune response. Anti-S1 and anti-RBD IgG are induced in both serum and BAL and IgA in BAL. Neutralizing antibodies are present in both serum and BAL. Little immunogenicity is observed with similar vaccines containing the RBD portion of the spike protein. The discussion section is well written.
1) The manuscript is not very systematic in its comparison of different vaccines. The vaccines compared differ in the location of immunization, the adjuvant used, and the vaccine schedule of the study. It would be better if they compare two vaccines differing in only one variable at a time to determine if the S1+CpG intranasal vaccine is really better than the same vaccine given subcutaneously over the same number and schedule of doses. They should justify their use of different adjuvants and schedules for comparison.
2) The authors measure only the antibody response. Although they note that other intranasal vaccines induce cellular responses, they do not show any results of cytokine measurements or IgG antibody isotype measurements to determine the amount and type of T cell response. Such measurements should be included.
3) The authors boost every week. It is possible that they are boosting before the response to the prior boost has resolved and therefore blunt the response. They might get a better response boosting every 2 weeks or so. They should try this or justify the weekly boost.
4) In line 93, the authors state that the concentration of the recombinant proteins are 0.5 microgram per ml. It is not possible to generate a vaccine containing 10 micrograms in 30 microliters from this solution. This is a mistake in the concentrations.
5) OD measurements go up to 3 (0.1 % transmittance), which is not accurate. For such samples, they should test a dilution and calculate the OD.
Comments on the Quality of English Language1) Line 28 and 258 should read “every other group.”
2) Line 37 remove “a”
Author Response
1) The manuscript is not very systematic in its comparison of different vaccines. The vaccines compared differ in the location of immunization, the adjuvant used, and the vaccine schedule of the study. It would be better if they compare two vaccines differing in only one variable at a time to determine if the S1+CpG intranasal vaccine is really better than the same vaccine given subcutaneously over the same number and schedule of doses. They should justify their use of different adjuvants and schedules for comparison.
Response: In terms of comparative research, the reviewer's point is scientifically correct. If the comparison is between theefficiency of intranasal and subcutaneous injection, we could consider matching the dosage as well as the timing and frequency of administration for both methods. However, when considering the primary objective of clinical application, the intranasal method is relatively noninvasive and allows for repeated administration by the patient. On the other hand, administering vaccines viainjection carries the risk of pain for non-immunized individuals and requires a medical professional to perform the injection.Therefore, in this study, we compared and contrasted the two different administration methods from a practical perspective, and developed a more clinically acceptable method in terms of dosage and frequency for both methods. By incorporating and expanding on our innovative approach, we believe that the promising results achieved with the combination of S1 and CpG administered intranasally deserve a more practical administration method report.
2) The authors measure only the antibody response. Although they note that other intranasal vaccines induce cellular responses, they do not show any results of cytokine measurements or IgG antibody isotype measurements to determine the amount and type of T cell response. Such measurements should be included.
Response: We agree with the point of view of the reviewer. Consequently, we have updated the limitations paragraph in the manuscript as follows:
“Certain limitations were noted in this study. First, we evaluated humoral immunity via quantification of antibody titers but did not analyze cell-mediated immunity, which is also crucial in SARS-CoV-2 immunity. Therefore, assessing cell-mediated immunity, such as cytokine production, is necessary.” (Page 12; Lines 382–384)
3) The authors boost every week. It is possible that they are boosting before the response to the prior boost has resolved and therefore blunt the response. They might get a better response boosting every 2 weeks or so. They should try this or justify the weekly boost.
Response: Although we agree with the reviewer’s point of view, considering the main objective of clinical application, we have shown in the macaque model that the third or fourth dose of the adjuvanted subunit vaccine provides full protection against viral challenge. Specifically, the mucosal boost induced local respiratory mucosal protection and greatly enhanced or synergized with systemic immunity to rapidly eliminate viruses in the nasal cavity and prevent viral infection. Therefore, we designed the experiment with a one-week interval for boosting, emphasizing the importance of administering the vaccine four times. We have included the rationale for using adjuvants in the Introduction and Discussion sections as follows:
"In the macaque model, our previous research has demonstrated that administering a third or fourth dose of an adjuvanted subunit vaccine provides complete protection against viral challenge. Therefore, the experimental design for this study is based on administering a weekly booster dose." (Page 10 Lines 291–294)
4) In line 93, the authors state that the concentration of the recombinant proteins are 0.5 microgram per ml. It is not possible to generate a vaccine containing 10 micrograms in 30 microliters from this solution. This is a mistake in the concentrations.
Response: We apologize for this mistake, which has been rectified on Page 5, Line 93 as follows:
“Recombinant RBD protein (0.5 mg/mL) and S1 protein (0.5 mg/mL) were combined with CpG-ODN (hereafter CpG) or alum.”
5) OD measurements go up to 3 (0.1 % transmittance), which is not accurate. For such samples, they should test a dilution and calculate the OD.
Response: We thank the reviewers for their valuable comments and suggestions. We believe that the upper limit specified for the microplate reader is an absolute value, approximately OD 2.5, which is linear. Fortunately, since there are only three samples in Figure 3a) and two samples each in Figure 4a) and b) that exceed the value of 2.5, it would not impact the statistical results. Therefore, although we agree with the reviewer's point of view, due to the absence of a reference antibody titer, we were forced to conduct the measurements in the same environment and determine the increase in antibody titer based on OD values. In the case of dilution, the guarantee of a linear relationship between the absolute value of OD and the dilution ratio no longer exists, making the OD an unreliable measure. Fortunately, the seven samples with OD > 2.5, where linearity is not guaranteed, were the same ones shown in Figures 3 and 4. As depicted in the figures, the statistical outcomes were not affected. As a precautionary measure, we have added a comment about the samples with OD > 2.5 in the legends of Figures 3 and 4."
- Line 28 and 258 should read “every other group.”
Response: As suggested, we have corrected the term to “every other group”.
- Line 37 remove “a”
Response: As suggested, we have removed the “a”.
Round 2
Reviewer 2 Report (New Reviewer)
Comments and Suggestions for Authors
Revisions are acceptable.
This manuscript is a resubmission of an earlier submission. The following is a list of the peer review reports and author responses from that submission.
Round 1
Reviewer 1 Report
Comments and Suggestions for Authors
This research paper focus on CpG as an adjuvant for intranasally administered covid vaccine. It compared the said vaccine against subcutaneously injected vaccine with alum as an adjuvant. The authors have clearly explained how first two experiments resulted in negative outcomes and later they changed the design to for third experiment. The paper does poor job at designing experiments, measuring enough indicators to reach to conclusion and overall defining their purpose of these experiments. The experiments are designed without representative negative or positive control. For example, using 1000 ug of alum subcutaneously instead of appropriate amount of alum intramuscularly would be a true positive control considering recombinant protein-based vaccines for covid are injected intramuscularly. The dose of the alum is too high and no rational is given for that. Throughout the study, no rational is given for choosing dose, dosing schedule and control. For example, why author chose same dose of RBD and S1 protein and not same molecularly equivalent dose of RBD and S1 protein so amount of RBD administered between different studies is constant. By choosing the same dose of S1 protein, actually less amount of RBD is delivered. This is fine as the antibody response depends on how protein is presented and thus larger protein like S1 could be better than RBD in this case. However, no rational or detailed discussion is done on such topics. The author did not measure any indicators of innate immune response like cytokine level. Even adaptive response indicator measurement was restricted to IgG and IgA, whereas it is more common to also check IgG1, IgG2 responses in addition to know Th1 or Th2 bias. Overall, the paper could have just three figures, one per each study but it extends it to six figures without actually adding more data. Overall, this paper needs major changes in terms of design of experiment and need to obtain much more data to come to a certain conclusion.
Comments on the Quality of English Language
The article does not have issues with spelling or grammar but it does some time unknowingly exaggerate the meaning. For example, "Aluminum hydroxide, a typical clinical adjuvant, is 59 cytotoxic and causes cell necrosis [6]." It would have been good to use the word inflammatory or locally cytotoxic there.
Author Response
Thank you for pointing this out. We agree with this comment. Therefore, I updated text in the manuscript if necessary.

Reviewer 2 Report
Comments and Suggestions for Authors
Muranishi and colleagues report here the comparison of intranasal and subcutaneous immunization of SARS-CoV-2 spike related proteins. They showed that intranasal immunization can induce both serum-specific IgG titers and IgA titers in the alveolar lavage fluid.
Overall, the study and manuscript is very straightforward. It provides new evidence supporting the development of intranasal vaccination against respiratory viruses.
Please find below my comments:
Major remarks:
1. The authors should indicate why CpG-ODN and Alum are used as adjuvants in this study.
2. In the introduction, the authors emphasize the importance of intranasal delivered vaccines against respiratory viral infection. Considering few such SARS-CoV-2 vaccines have been approved, the author may give some successful examples from other respiratory virus.
3. In this study, the RBD is a commercial one purchased from Sion Biological, while the spike S1 protein is made in house. The commercial protein should be endotoxin-free. Did the author detect the endotoxin level of S1 protein? In vivo, endotoxins may induce an inflammatory response in animal studies.
4. The author should summarize different experiments with a name rather than using “1st, 2nd, 3rd experiment”.
5. In the method, 1,000 ug CpG was used. But the authors showed “10 ug CpG” in the Result part. The authors should make it clear and consistent.
6. What is the conclusion and what are the lessons that we can learn from the first experiment? The author may make a brief summary of 1st experiment and explain why and how they would like to design and start their 2nd experiment. The author can add some description similar to line 221-223.
7. Did the author measure the anti-RBD IgG titer in BAL in 2nd experiment? If yes, they should include the data in the manuscript.
8. It will be better for the readers to understand the design of the study if the schematic diagram of animal experiment can be moved to relevant Figure rather than putting them in the Method part.
9. In 3rd experiment, “S1 (alone) s.c.” group should be included to make the control group complete; otherwise, the conclusion in line 235-236 will not be convincing since different adjuvant was used and antigens were delivered in different ways.
10. Most of the ELISA data is the raw OD value based on the diluted 1:1000 serum, which is only one concentration. Typically, serum titer is calculated based on serially diluted samples. The authors should remove the “titer” from the Y axis title.
11. The author would like to show the advantage of intranasal vaccination in eliciting local immune response, especially the IgA titer in BAL. However, the final key data of this study, neutralization data, is generated using serum sample, which did not contain Spike-specific IgA antibody. I am also surprised to see the data from Figure 4 Figure 6, showing that intranasal immunization induced stronger serum IgG response compared to subcutaneous immunization. Typically, subcutaneous delivery of antigen is potent in eliciting systemic humoral immune response, like serum IgG. Could the author explain the phenomenon in this study and provide related published papers?
12. Strictly speaking, the assay used in Figure 6 is ACE2 receptor blocking assay. The typical neutralization assay should be performed in pseudovirus or SARS-CoV-2 virus. Besides, the author should indicate the average value the inhibition data in Figure 6. The averages of “S1 + CpG i.n.” and “S1 + CpG s.c.” look similar even though they are statistically significant.
13. Did the author detect the so-called “neutralization titer” of sample in BAL? This is an essential data to show the advantage of intranasal vaccination in inducing local humoral immune response (highlight of this study) including neutralizing antibodies.
14. In this study, the author only focus on humoral response. Nevertheless, cell-mediated immunity also play an important role in preventing viral infection. It will make this study more complete, interesting and attractive if the authors can include the data explaining the cellular immunity.
15. Can adjuvant Alum delivered i.n.?
Minor remarks:
1. Reference for line 41-42 “… is considered weak”
2. 3. typo line 57 ”illicit”
3. typo line 192 “)”
4. Reference for line 285-287. Or incorrect position of the reference 16?
Comments on the Quality of English LanguageModerate editing of English language required
Author Response

(The authors gave the same response as above.)

Reviewer 3 Report
Comments and Suggestions for Authors
A well-designed and properly conducted study, however, due to the subsidence of the COVID-19 pandemic, interest in the results may be limited.
Author Response
We would like to thank the reviewers for their time and valuable comments and suggestions.
Reviewer 4 Report
Comments and Suggestions for Authors
Authors have reported the antibody response following intranasal administration of SARS-CoV-2 spike protein-CpG oligonucleotide vaccine authors should correct the following
1. How was the dose calculated? Give reason
2. Advantages of nasal administration should be given.
3. Recent references for sars cov 2 should be given authors may go through 10.1002/jmv.27936
4. Has the authors compare with the other vaccine available . How it was beneficial?
5. Does this vaccine produce any toxicity.
6. Check for english grammar and spelling errors.
7. Recent refences should be provided with proper format
Author Response

(The authors gave the same response as above.)

Round 2
Reviewer 1 Report
Comments and Suggestions for Authors
The edits and comments made in response to my review is not sufficient. Most of the questions are not addressed. It will take a good amount of experiment planning and conducting those experiments to publish a paper. While I appreciate the amount of work that the authors did to put this article together, it does not match the scientific rigor required for a study to be conducted and published. In fact, it is not the actual amount of the work but how scientifically sound it is that matters, and thus I am not in favor of publishing this paper.
Author Response
[Response] Thank you for your valuable time and peer review. I would also like to rework my research plan to focus on S1 protein instead of RBD. We will re-design the experiment.
We are immunizing mice to add a control group with S1 protein alone, which will take about 50 days. We are planning to measure neutralizing antibodies in BAL at the same time.
Reviewer 2 Report
Comments and Suggestions for Authors
In the revised version of the manuscript, it is a pity that the author did not take the suggestions from the reviewer to add the necessary data, making the work more convincing.
In this study, the authors would like showed the antibody response following intranasal immunization of spike protein. Mucosal immunity is the major response induced by intranasal vaccination, especially the IgA antibody response. Typically, only neutralizing antibody can contribute to prevent viral infection. In summary, neutralizing antibody in the respiratory tract will be the essential data for this paper. But such data is missing in this manuscript and the author did not perform this experiment even though I gave the suggestions in the “major remarks”. On the other hand, both IgG and IgA may be elicited in BAL. The author should also test the IgG antibody response in BAL.
It is important to have sufficient literary research before the author design and start the experiments. It is a common sense that proteins with small molecule weight usually show poor immunogenicity. Besides, there are already many publications indicating that RBD is poorly immunogenic because its molecular size is small for antigen presentation by antigen-presenting cells (APCs) (PMID: 34545190; PMCID: PMC9940465). The preliminary data in this study about RBD immunization is as expected. In this way, the author should design the study better by focusing on the S1 protein rather than spending too much effort on RBD.
Complete controls are indispensable for drawing a convincing conclusion. I suggested the author to add the “S1 (alone) s.c.” group in their 3rd experiment to have a better comparison. The author’s response is “Although a mild increase in antibody titer can be expected with subcutaneous administration of S1 protein alone, we chose to administer S1 alone intranasally to minimize the use of animals.” How can the result be expected without real generated data? If the outcome of the experiment can be estimated, what is the meaning of performing the experiment? According to author’s statement, it is unnecessary to perform the RBD immunization since the low immunogenicity of RBD is expected. The author refused to add the control data because they would like to minimize the use of animals. However, the fact is that much more animals were used unnecessarily just to test the immunogenicity of RBD, which is already proven to be poorly immunogenic.
For my comment in last-round comments-to-author “Most of the ELISA data is the raw OD value based on the diluted 1:1000 serum, which is only one concentration. Typically, serum titer is calculated based on serially diluted samples. The authors should remove the “titer” from the Y axis title.” The author did not change as suggested by me. I feel confused why they did not correct them.
It seems that the author did not carefully check their manuscript and “author’s response” since I find new mistakes in both material. In Figure 4, one of the group name is missing in the X-axis, which should be “S1 + Alum s.c.”. In author’s response, the question of point 14 contains the answer for point 13, which can be correct if the author can really check the material before submitting. The above mistakes leave an impression for the reviewer that they did not take the revision very seriously.
Comments on the Quality of English LanguageModerate editing of English language required
Author Response
In the revised version of the manuscript, it is a pity that the author did not take the suggestions from the reviewer to add the necessary data, making the work more convincing.
In this study, the authors would like showed the antibody response following intranasal immunization of spike protein. Mucosal immunity is the major response induced by intranasal vaccination, especially the IgA antibody response. Typically, only neutralizing antibody can contribute to prevent viral infection. In summary, neutralizing antibody in the respiratory tract will be the essential data for this paper. But such data is missing in this manuscript and the author did not perform this experiment even though I gave the suggestions in the “major remarks”. On the other hand, both IgG and IgA may be elicited in BAL. The author should also test the IgG antibody response in BAL.
[Response] I would like to submit the results of neutralizing antibodies in the respiratory tract. We would also like to examine the IgG antibody response in BAL, as both IgG and IgA may be elicited in BAL. Please give us some time as it takes about 50 days for immunization.
It is important to have sufficient literary research before the author design and start the experiments. It is a common sense that proteins with small molecule weight usually show poor immunogenicity. Besides, there are already many publications indicating that RBD is poorly immunogenic because its molecular size is small for antigen presentation by antigen-presenting cells (APCs) (PMID: 34545190; PMCID: PMC9940465). The preliminary data in this study about RBD immunization is as expected. In this way, the author should design the study better by focusing on the S1 protein rather than spending too much effort on RBD.
[Response] I would like to redesign my research with a focus on the S1 protein.
Complete controls are indispensable for drawing a convincing conclusion. I suggested the author to add the “S1 (alone) s.c.” group in their 3rd experiment to have a better comparison. The author’s response is “Although a mild increase in antibody titer can be expected with subcutaneous administration of S1 protein alone, we chose to administer S1 alone intranasally to minimize the use of animals.” How can the result be expected without real generated data? If the outcome of the experiment can be estimated, what is the meaning of performing the experiment? According to author’s statement, it is unnecessary to perform the RBD immunization since the low immunogenicity of RBD is expected. The author refused to add the control data because they would like to minimize the use of animals. However, the fact is that much more animals were used unnecessarily just to test the immunogenicity of RBD, which is already proven to be poorly immunogenic.
[Response] Thank you for pointing this out, we would like to add subcutaneous control of S1 protein only. We have already started immunization, so could you please give us a little more time?
For my comment in last-round comments-to-author “Most of the ELISA data is the raw OD value based on the diluted 1:1000 serum, which is only one concentration. Typically, serum titer is calculated based on serially diluted samples. The authors should remove the “titer” from the Y axis title.” The author did not change as suggested by me. I feel confused why they did not correct them.
[Response] I am very sorry, I would like to remove remove the “titer” from the Y axis title. We would like to add a subcutaneous control for S1 protein only and submit a revised version of the relevant figure next time.
It seems that the author did not carefully check their manuscript and “author’s response” since I find new mistakes in both material. In Figure 4, one of the group name is missing in the X-axis, which should be “S1 + Alum s.c.”. In author’s response, the question of point 14 contains the answer for point 13, which can be correct if the author can really check the material before submitting. The above mistakes leave an impression for the reviewer that they did not take the revision very seriously.
[Response] We are very sorry. In Figure 4, we have corrected this by adding "S1 + Alum s.c." to one of the group names for the X axis. We would like to add a subcutaneous control for S1 protein only and submit a revised version of the relevant figure next time.
Reviewer 4 Report
Comments and Suggestions for Authors
Comments incorporated
Author Response
Reviewer 4
- How was the dose calculated? Give reason
[Response] The dosage was determined based on the results of previous studies. This has been added to the Methods section, along with the relevant literature review. Page 3, Lines 102–103: “RBD (10 µg)-CpG (1,000 µg) and CpG (1,000 µg) alone were nasally administered, as previously described [9]. ”
- Advantages of nasal administration should be given.
[Response] We added in the Introduction section that the dosage was determined based on previous studies along with a literature review.
Page 2, Lines 55–57: “Intranasal administration of attenuated live influenza vaccine is now an important part of the influenza vaccination strategy, and clinical trials for intranasal pertussis vaccine are underway [7]. Therefore, generating a nasal administrative vaccine that enhances the immunity of the respiratory tract mucosa, while being less invasive with fewer side effects, is imperative.”
Page 11, Line 384–388: “Compared to subcutaneous administration of vaccines, nasal administration of vaccines is beneficial for immune responses and many other practical aspects, as it does not require a sterile environment, can be administered safely to children of all ages, people themselves can easily administer booster vaccinations, and there exists no possibility of inappropriate use of needles or risk of infectious disease transmission [43].”
- Recent references for sars cov 2 should be given authors may go through 10.1002/jmv.27936
[Response] We referred to 10.1002/jmv.27936 and added in the Introduction section that the dosage in the past literature was referenced, along with the literature.
Page 1, Line 38–39: Furthermore, SARS-CoV-2 has evolved with several genomic characteristics, varying severity of infection, and evading immunity [2].”
- Has the authors compare with the other vaccine available . How it was beneficial?
[Response] We have added the used reason of adjuvants in the discussion and conclusion sections as follows:
Page 9, Line 272–274 and 319–320: “In this study, intranasal administration of S1-CpG increased serum IgG and BAL IgA antibody titers against S1 and RBD, whereas intranasal administration of RBD-CpG did not increase the titer.” and “In this study, alum was used as an adjuvant for subcutaneous administration for comparison with CpG-ODNs.”
Page 11, Line 399–402: “The nasal administration of the S1 protein vaccine with CpG as an adjuvant is useful and a potential vaccine candidate. We conclude that the RBD portion of the S1 protein alone was insufficient to induce immunity regardless of the route of administration, with alum or CpG as adjuvants.”
- Does this vaccine produce any toxicity.
[Response] We have added the reason in the discussion as follows:
Page 9, Line 347–349: “Most side effects of a subunit recombinant spike protein vaccine combined with Advax-CpG55.2™ adjuvant were mild, with complete recovery in a short time [37].”
- Check for english grammar and spelling errors.
[Response] English grammar and spelling errors were checked by all authors. We also sought the help of native English speakers.
- Recent refences should be provided with proper format
[Response] References have been formatted as per the target journal guidelines.